# Advantages and Limitations of Animal Schizophrenia Models

**DOI:** 10.3390/ijms23115968

**Published:** 2022-05-25

**Authors:** Magdalena Białoń, Agnieszka Wąsik

**Affiliations:** Department of Neurochemistry, Maj Institute of Pharmacology, Polish Academy of Sciences, 31-343 Cracow, Poland; bialon@if-pan.krakow.pl

**Keywords:** animal models of schizophrenia, amphetamine, phencyclidine (PCP), ketamine, dizocilpine (MK-801), neurotrophic factor neuregulin 1 (NRG1), disrupted-in-schizophrenia 1 (DISC-1) gene, neonatal ventral hippocampal lesion (NVHL), maternal immune activation (MIA), methylazoxymethanol acetate (MAM)

## Abstract

Mental illness modeling is still a major challenge for scientists. Animal models of schizophrenia are essential to gain a better understanding of the disease etiopathology and mechanism of action of currently used antipsychotic drugs and help in the search for new and more effective therapies. We can distinguish among pharmacological, genetic, and neurodevelopmental models offering various neuroanatomical disorders and a different spectrum of symptoms of schizophrenia. Modeling schizophrenia is based on inducing damage or changes in the activity of relevant regions in the rodent brain (mainly the prefrontal cortex and hippocampus). Such artificially induced dysfunctions approximately correspond to the lesions found in patients with schizophrenia. However, notably, animal models of mental illness have numerous limitations and never fully reflect the disease state observed in humans.

## 1. Introduction

Schizophrenia is a chronic, severe mental illness with a wide range of clinical and biological manifestations [1]. The prevalence of schizophrenia in the population is approximately 1% [2]; however, the prevalence significantly varies across places and social groups [3]. The first features of the illness manifest in late adolescence or early adulthood [4], and the features, course, prognosis, and treatment outcome may differ across individuals. In schizophrenia, the following three groups of symptoms are commonly distinguished: positive, negative, and cognitive symptoms (detailed in Figure 1) [4,5].

The pathogenesis of schizophrenia remains unclear, although it has been widely investigated for decades. However, regarding previous research examining the factors responsible for developing the illness, the following three groups can be distinguished: neurodevelopmental, genetic, and environmental factors.

Currently used antipsychotic drugs do not meet the expectations of doctors and patients because they do not fully reverse all three types of symptoms characteristic of schizophrenia. Moreover, up to 40% patients do not respond adequately to treatment with antipsychotic [6]. In addition, typical antipsychotics display strong extrapyramidal side effects, which include parkinsonism and dystonia [7], while the atypical antipsychotic treatment is related to metabolic syndrome, resulting in type II diabetes and/or hyperlipidosis [8]. To search for new, effective drugs, it is necessary to use appropriate animal models that mimic the disorders appearing in patients with schizophrenia. Moreover, animal models provide a better understanding of the etiopathology of the disease and the mechanisms involved in currently used therapies.

Mental illness modeling is a major challenge for scientists because laboratory animals (most of which are rodents) do not exhibit higher mental functions. Therefore, it is difficult to determine the mental state of the tested animal. Modeling such diseases is based on inducing damage or changes in the activity of relevant regions in the rodent brain. Such artificially induced dysfunctions approximately correspond to the lesions found in patients with schizophrenia. However, notably, animal models of mental illness never fully reflect the disease state observed in humans. Various types of animal models of schizophrenia are used in basic research. These models include pharmacological models (based on dopamine (DA) system hyperfunction or glutamate (GLU) system hypofunction), genetic models (based on disorders in genes related to schizophrenia), and neurodevelopmental models (involving damage to relevant brain regions during the fetal or neonatal period of rodents).

## 2. Pharmacological Models

Pharmacological animal models of schizophrenia seem to be the most popular and are based on the hyperactivity of DAergic brain pathways, primarily midbrain DA projections to limbic regions [9,10], or the hypofunction of the N-methyl-D-aspartate (NMDA) receptor, which is expressed in most cells in the central nervous system [11]. 

### 2.1. Dopaminergic Hyperfunction Model

It has been noted that DA enhancers, such as amphetamine, administered to healthy volunteers cause a psychosis state resembling the psychoses observed in amphetamine addicts [12]. In animals, the administration of escalating, intermittent doses of amphetamine induces a sensitized state resembling some neurochemical and behavioral features relevant to the positive symptoms of schizophrenia [13], which is called ‘amphetamine sensitization’. The strength of sensitization may depend on various factors, such as the number of and interval between treatments, dose, sex, age, and genetics [14]. Research involving rodents using an amphetamine-induced schizophrenia model showed hyperlocomotion [15], spatial working memory impairments [15,16], sensorimotor gating deficits measured in prepulse inhibition (PPI) [17,18], and latent inhibition deficits [18]. However, the amphetamine-based model in rodents does not show impairments in social tasks, suggesting no effect on social behavior, which is thought to be pertinent to negative schizophrenia symptoms [16]. In addition to the increased release of DA in the nucleus accumbens and striatum, repeated exposure to amphetamine induces many changes in brain morphology and physiology, such as the upregulation of α-amino-3-hydroxy-5-methyl-4-isoxazolepropionic acid (AMPA) receptors, altered NMDA receptor expression, and altered dendritic morphology [19,20]. In addition, amphetamine-induced sensitization alters functioning cholinergic (ACh-ergic) basal neurons, projecting to cortex [21], and it has been suggested that the deficits in attentional function induced by amphetamine may be connected to the dysregulation of ACh-ergic activity [22]. On a molecular level, DAergic enhancers, such as amphetamine, may lead to increased tumor necrosis factor α (TNFα) and malondialdehyde (MDA) levels, suggesting ongoing increased inflammatory and oxidative processes [16]. Similar effects were observed in zebrafish, in which amphetamine was able to alter the oxidative status [23]. Proteomic analyses conducted by Wearne et al. [24] identified 96 proteins that are differentially expressed in the prefrontal cortex in a rat methamphetamine-based model of sensitization; of these differentially expressed proteins, 20% are connected to schizophrenia pathogenesis and linked to synaptic regulation, protein phosphatase signaling, mitochondrial function, and impairments in the inhibitory gamma-aminobutyric acid (GABA) system. Animal research using DA enhancers as a schizophrenia model mainly focuses on behavioral changes; thus, data showing oxidative, inflammatory, or neurotransmitter changes are limited.

**Advantages**: The generation of an amphetamine-induced schizophrenia model is a very simple, inexpensive, and not time-consuming process and consists only of repeated escalating doses of amphetamine. Moreover, psychomotor hyperactivity and memory deficits are observed immediately after the administration of amphetamines. Thus, the amphetamine model is useful for studying some effects of a manipulation in a short period that may be related to schizophrenia.

**Limitations**: This model mainly causes deficits associated with the positive symptoms of schizophrenia and some cognitive deficits, especially those related to attention. Thus, this model does not fully reflect the complexity of schizophrenia, and the model provides limited value in understanding the etiopathology of the illness. Moreover, special care should be taken because after amphetamine administration, animals are very excitable and may be aggressive.

### 2.2. Dopaminergic Hypofunction Model

Negative symptoms of the illness may be modeled by inducing frontal hypodopaminergic effect, which is representative of schizophrenia symptoms, such as poor motivation [25], as people diagnosed with schizophrenia show a tendency to make decisions linked to lower effort [26]. Injection of 6-OHDA into rats’ mPFC leads to increased DA levels in striatum and nucleus accumbens [27], which points to the theory of increased DA signaling in subcortical regions and diminished one in frontal parts of the brain in schizophrenic subjects [28].

In behavioral tests, frontal DA hypofunction may lead to spatial memory impairment [29] and decreased social behavior [30,31], however, with no effect of lesion on performance in open-field or novel investigation tests [31]. On the other hand, Rezvani and coworkers [32] showed increased locomotor activity in 6-OHDA-lesioned animals and, what is interesting, the improvement of performance in an eight-arm radial maze. Mice with a depleted DA level in nucleus accumbens choose the lower effort options without the desire to reach for a valued result [26], showing disturbed motivation-related neuronal processes.

**Advantages**: The advantage of lesion-induced changes is rather limited, but this model is useful for modeling the specific aspects of negative symptoms, such as motivation and effort-related behavior. The restricted behavioral results of 6-OHDA lesions are useful for new therapy screening, allowing for evaluating their contribution for the improvement of the specific manifestation of the illness.

**Limitations**: However, to our knowledge, the model is rarely used to model symptoms of schizophrenia and is rather applied in studies linked to the role of DA-driven and motivation/effort-related behavior; thus the number of previous research studies using 6-OHDA lesions in animals, specifically as a schizophrenia model, is limited. Available data making use of the 6-OHDA lesion model in the context of schizophrenia is disputatious due to differing results obtained mainly from behavioral studies. 

### 2.3. Serotonergic Model

Another compound used to model schizophrenia-like behavior is 2,5-dimethoxy-4-iodoamphetamine (DOI), which is an amphetamine substitute and acts as a 5-HT2A/2C agonist. Both types of serotonergic receptors (5-HT2A and 5-HT2C) are localized to the same neurons within the medial prefrontal cortex (mPFC), which regulates executive function, decision making, and reward-guided learning and memory processes [33]. 

Activation of 5-HT2A/2C receptors by DOI significantly alters glutamate release and glutamatergic pyramidal neuron activity in mPFC. Moreover, DOI increases 5-HT, DA, and ACh release in the rodent’s PFC [34]. Administration of DOI induces head-twitch response (HTR), resembling positive schizophrenia symptoms (particularly hallucinations), and this effect is blocked by antipsychotic agents [34]. Moreover, DOI disrupts PPI [35,36], however, without changes in locomotor activity [37]. On a molecular level, DOI induces an increase in the frequency of spontaneous excitatory postsynaptic currents (sEPSCs) in pyramidal neurons [38,39]. Additionally, measured in microdialysis, DOI produces an increase in glutamate release in mPFC [40], a characteristic feature of other pharmacological schizophrenia models. Worth mentioning is that the administration of an NMDA receptor antagonist, MK-801, enhances DOI-induced head twitches and locomotor activity, suggesting synergic action of these agents [41].

**Advantages**: Inducing HTR by DOI administration is a simple, fast, and cheap procedure. Moreover, the DOI-induced behavior changes are suitable model due to the fact that HTR is characterized as an easily quantifiable, reliable response with low variability between and within the subjects [34], and useful as a screening tool for assessing the potential of antipsychotics agents to attenuate positive symptoms.

**Limitations:** Based on our observations, DOI is rarely used to study spontaneous locomotor activity (or does not exert the effect of increased activity, as mentioned above) or to mimic symptoms including sociability or memory; thus its action is limited to modeling positive manifestation of the illness exclusively. Simultaneously, DOI is most often used as an additional compound, apart from other schizophrenia model used in the experiment, due to its effect limited to the one behavioral manifestation.

### 2.4. Glutamatergic Hypofunction Models

The schizophrenia model of GLUergic hypofunction started receiving attention following recurring observations that injections of NMDA receptor antagonists at low doses, such as ketamine or phencyclidine (PCP), induce psychosis-like states and cause negative symptoms in healthy subjects, resembling these symptoms in schizophrenia patients [42]. Moreover, NMDA receptor antagonists deteriorate existing symptoms in schizophrenic individuals [43].

PCP is a psychomimetic drug that acts as a noncompetitive NMDA receptor antagonist [44]. Presenting a wide range of central nervous system manifestations, PCP administration serves as a well-established animal schizophrenia model [44].

Both acute and subchronic PCP injections at different doses cause various behavioral deficits in animals relevant to human schizophrenia symptoms, such as impairments in memory and learning [45,46], sociability [47,48], locomotor activity [49], and sensorimotor gating measured in PPI [50]. Notably, some studies have failed to confirm important features of schizophrenia-like behavior in rodents [47], indicating that the effects of PCP in behavioral studies may differ depending on the dose used and the administration paradigm. Several studies have investigated differences in results obtained in acute and subchronic PCP administration paradigms. Castañé and coworkers [51] showed that both acute and subchronic injections of PCP change animals’ locomotor response; however, only a subchronic administration pattern impairs working memory and changes tyrosine hydroxylase (TH) mRNA expression in the VTA [51]. Thomson et al. showed that different aspects of mental processes are disturbed in different paradigms—acute NMDA receptor blockade by PCP administration leads to impaired inhibitory control, whereas the chronic one causes processing speed deficits [52].

The effects of PCP at the molecular level involve electrophysiological, structural, functional, neurotransmitter, and protein alterations. First, in [53], PCP induced an increase in DA efflux in the medial prefrontal cortex and, in [54,55], significantly enhanced the GLU, DA, noradrenaline (NA), and serotonin (5-HT) extracellular levels in the prefrontal cortex with a decrease in GABA efflux, confirming the alteration in frontal excitatory/inhibitory circuits. Moreover, PCP administration affects brain-derived neurotrophic factor/tropomyosin receptor kinase B (BDNF/TrkB) signaling, diminishing the brain levels of both BDNF and its receptor [56,57]. Reduced frontal and hippocampal nerve growth factor (NGF) levels were also observed in [58]. Another factor affected by PCP is parvalbumin (PV)-positive interneurons, which are important regulators of pyramidal activity. Some research has shown a decreased number of frontal PV+ cells in PCP-treated animals [50], suggesting altered inhibitory signaling in frontal regions. Electrophysiological studies involving PCP-treated animals show impaired long-term potentiation (LTP) in the medial prefrontal cortex [56] and hippocampus [59], a significant process underlying the basis of synaptic plasticity, learning, and memory. Moreover, PCP can activate the inflammatory response by enhancing the mRNA production of proinflammatory cytokines, such as interleukin-6 (IL-6), TNFα, and interleukin-1β (IL-1β) [60]. Additionally, PCP administration causes disrupted oxidative stress markers as it leads to decreased superoxide dismutase (SOD), catalase (CAT), and glutathione (GSH) levels and activity with a simultaneous increase in MDA levels [60], suggesting increased lipid peroxidation and an impaired defense system against oxidative damage.

Kaiser et al. [61] used microarray analysis and identified a changed expression in 477 genes in the prefrontal cortex of PCP-treated rats, confirming that PCP may also affect gene expression. These changes involved genes related to stress; inflammatory response, growth, and development; neural plasticity; and signal transduction, and these genes all seem to be disturbed in schizophrenia or are important for the risk of developing the illness. Worth mentioning is that chronic PCP treatment, compared with acute paradigm, shows more prominent and long-lasting molecular changes relevant to schizophrenia symptoms (e.g., different energy metabolism or biosynthetic processes) [62].

**Advantages**: The generation of the PCP schizophrenia model is a very simple, cheap, and not time-consuming process as it consists of only subchronic PCP injections. Psychomotor hyperactivity and psychosis-like behavior are observed immediately after the PCP administration. What is important is that PCP administration induces negative and cognitive symptoms as well. Moreover, different administration paradigms may be adapted to expected effects of NMDA receptor blockade (e.g., subchronic schedule to induce memory deficits or the acute one to produce hyperactivity or disinhibition of cortical neurotransmitters, such as DA, GLU, or 5-HT) [62]. This model is useful for studying the effects of a pharmacological intervention attenuating all types of schizophrenia-related symptoms. 

**Limitations**: This model provides limited value in explaining the etiopathology of the illness. Moreover, special care should be taken because after PCP administration, animals are very excited and may be aggressive.

**Ketamine** is another NMDA receptor antagonist. This compound is a derivative of PCP and is extensively used as an anesthetic agent in both human and veterinary medicine [63]. Some authors have reported deterioration of mental status in schizophrenic subjects after an injection of ketamine at a subanesthetic dose [43,64]. When injected into healthy volunteers, ketamine produced both negative and positive schizophrenia symptoms in [65]. Based on these observations, ketamine began to be used to model schizophrenia symptoms in animals. Studies mainly include rodent models; however, ketamine has also been studied in zebrafish [66] and monkeys [67], supporting our understanding of the molecular and behavioral basis of the illness.

The ketamine model of schizophrenia is thought to be applicable to explain the positive, negative, and cognitive symptoms; neurotransmitter disturbance; and other features characteristic of the illness [68]. Ketamine has been shown to produce a wide range of behavioral changes in not only rodents but also zebrafish, which are used to model schizophrenia-like behavior. Clinical and preclinical studies have reported that ketamine injections, both acute and subchronic, lead to memory impairments in humans [69], monkeys [70], and rodents [71,72,73]. Ketamine has also been shown to impair spontaneous alterations as a function of spatial working memory [74,75]. Moreover, ketamine can exert anxiety-like behavior [76,77], altered social behavior [78,79], hyperlocomotion [66,80], and impaired sensorimotor gating [81,82].

Ketamine has been reported to produce a wide range of molecular changes in animal brains. Ketamine affects neurotrophic factors, such as BDNF, that maintain brain cell viability and physiological neuronal communication [83]. Subchronic ketamine treatment was found to decrease the number of BDNF-positive cells in the posterior cingulate cortex [78], frontal cortex, hippocampus, and striatum [84] and BDNF protein levels in the amygdala, hippocampus, frontal cortex, and striatum [85,86]. Furthermore, ketamine injections in animals disturb other neurotrophic factors, such as NGF or neurotrophin-3 (NT-3) [83]. Ketamine administration is also responsible for inflammatory cytokine imbalance in schizophrenia animal models. Some authors have shown increased levels of proinflammatory IL-6 in the hippocampus and whole brain tissue in mice [86,87] and increased protein and mRNA levels of IL-6, IL-1β, and TNF-α in murine hippocampal tissue after a single injection of ketamine [88]. However, the data presented by Réus and coworkers [89] showed no significant changes in the levels of IL-1β, interleukin-10 (IL-10), IL-6, and TNF-α in several rat brain structures after subchronic ketamine administration.

Ketamine also reportedly disturbs the physiological redox balance by modulating oxidative stress. An injection of ketamine leads to a decrease in the levels of GSH, SOD, and CAT in the prefrontal, cortical, striatal, and hippocampal regions in rodent models [84,90,91]. The parameter used as a biomarker of oxidative stress is MDA, and ketamine injections have been shown to increase the MDA levels in rodent brains [84,92], suggesting ongoing oxidative stress reactions.

The other effects produced by ketamine administration include impairments in PV-expressing GABAergic neurons. Fujikawa et al. [82] showed reduced expression of PV+ neurons in the CA1 region of the hippocampus in ketamine-injected mice. Another study presented a lower density of PV+ neurons in the CA1–CA3 hippocampal regions and dentate gyrus [93] and generally reduced PV expression in both acute and subchronic ketamine administration models in hippocampal and prefrontal areas [93,94,95].

**Advantages**: The generation of a ketamine schizophrenia model is a very simple, inexpensive, and not time-consuming process and is based only on ketamine intraperitoneal injections. Ketamine produces negative, positive, and cognitive schizophrenia symptoms. Moreover, ketamine can exert anxiety-like behavior. Thus, this model mimics all behavioral disturbances observed in patients with schizophrenia. The ketamine model is useful for studying the effects of a manipulation that may be related to schizophrenia

**Limitations**: Researchers use different doses and regimens of ketamine administration; thus, they sometimes obtain divergent results (e.g., in some publications, ketamine administered in very low doses improves memory; however, ketamine given in higher doses may cause sedation), and the obtained results could be misinterpreted. This model provides limited value in understanding the etiopathology of schizophrenia.

**MK-801** (dizocilpine) is a noncompetitive NMDA receptor antagonist with greater inhibitory potency (IC50 4.1 ± 1.6 nM) than ketamine and PCP (IC50 508.5 ± 30.1 nM and IC50 91 ± 1.3 nM, respectively) [96]; therefore, MK-801 is more relevant for inducing schizophrenia-related behavioral and neurochemical changes in animal studies [97]. MK-801 is commonly used in mice and rats [98,99]; however, it is less often found in zebrafish [100] and monkey [101] studies. In rodents, the behavioral effects of MK-801 include working perturbations [102,103], hyperlocomotion [103,104], deficits in PPI [102,104], lowered sociability [103,105,106], and fear conditioning perturbations [106,107]. Notably, the data presenting MK-801-induced anxiety behavior seem to be rather inconsistent; some studies show the anxiogenic effects of MK-801 administration [105], while other studies present undisturbed anxiety-related behavior [104,108]. Additionally, Pınar and coworkers [109] showed that the administration of MK-801 during the neonatal period in mice may even reduce anxiety-like states. Interesting results were presented by Güneri et al. [110]. These authors showed that subchronic exposure to MK-801 may lead to antinociception in response to thermal and mechanical stimuli, which may contribute to developing research regarding the connection between schizophrenia and patients’ decreased pain sensitivity [111]. The increased locomotor activity in animals treated with MK-801 is linked to enhanced DAergic transmission in the striatum and nucleus accumbens [112,113], and increased DA levels were also observed in the prefrontal cortex in [114,115]. Increased subcortical DA release and related behavior are thought to serve as an experimental model of positive schizophrenia symptoms [116]. Further studies have shown that injection of MK-801 also leads to an increase in GLU efflux in the hippocampus and medial prefrontal cortex [117,118,119]. Regarding the role of the GLUergic system in schizophrenia pathophysiology, excessive GLU efflux in regions such as the prefrontal cortex and hippocampus is believed to produce symptoms of the illness, mainly cognitive symptoms [120]. Following GLUergic system disruption in schizophrenia, GABAergic dysfunction also occurs in animal models. Previous studies have shown decreased GABA and GABA-related protein and enzyme (such as PV and glutamate decarboxylase (GAD67)) levels in the prefrontal cortex and hippocampus of MK-801-treated animals [121,122]. Moreover, exposure to MK-801 in adolescent animals leads to long-term excitatory/inhibitory imbalance in the hippocampus and prefrontal cortex, resulting in reduced functional capacity and behavioral control [123,124]. Taken together, disrupted functioning of excitatory/inhibitory systems may lead to a broad spectrum of manifestations characteristic of schizophrenia (e.g., by the disinhibition of pyramidal neuron activity). MK-801 administration also affects neurotrophic factors, such as BDNF, which is responsible for LTP and long-term depression (LTD) and is involved in memory and learning processes in the hippocampus [125,126], which may explain, at least partially, the memory disturbances observed in animal schizophrenia models. However, previous studies have shown rather contrary results in terms of the BDNF levels after MK-801 treatment. Regarding the role of BDNF in memory/learning processes, MK-801 is expected to diminish the BDNF levels, and in some studies, it does, mainly in the hippocampus [127,128], but other studies have reported upregulated levels, including in vitro studies [129,130], which may suggest a response to the toxic action of MK-801 and/or action of compensatory mechanisms.

As mentioned above, pharmacologically induced NMDA receptor hypofunction leads to excessive extracellular GLU release, which causes excitotoxicity, leading to neuronal dysfunction, degeneration, and even neuronal death [131,132]. These effects may result in an increased cellular oxidative stress response [133], which is a widely studied feature of schizophrenia [134,135]. MK-801 has been shown to induce oxidative stress, including elevated MDA levels and altered antioxidant enzyme activity, in several brain regions in rodents, such as the prefrontal cortex, hippocampus, retrosplenial/posterior cingulate cortex [119,136,137], and cultured neurons and glial cells [138,139]. Additionally, MK-801 decreases GSH, a main antioxidative compound, leading to a reduced capacity to protect brain cells against oxidative stress [140,141].

**Advantages**: MK-801 serves as a relatively cheap and easy-to-perform method to develop schizophrenia symptoms in rodents or zebrafish. Similar to ketamine and PCP, pharmacological interventions may be performed in a short period before testing (e.g., 1 h), which reduces the costs of housing animals. Similar to other NMDA receptor antagonists and in contrast to DA enhancers, the MK-801 model may produce both positive and negative symptoms [142], showing efficacy in modeling the symptoms of illness in various behavioral tests and resulting in molecular changes characteristic of NMDA receptor antagonism.

**Limitations**: Similar to other pharmacological tools, the administration of MK-801 mostly resembles acute psychosis rather than a permanent state of the illness. Notably, MK-801 and other compounds are used to study schizophrenia-related behavior and molecular changes after acute or subchronic drug administration, whereas schizophrenia is a chronic disease with manifestations starting at puberty or shortly after this time [143]. Therefore, pharmacological manipulations do not mimic the pathogenesis of schizophrenia efficiently. These manipulations may resemble the symptoms; however, their ability to model the course of the development of the illness is rather limited [144]. Moreover, MK-801 is widely studied as a therapeutic agent mainly due to its anticonvulsant and neuroprotective (e.g., in stroke) properties [145]; therefore, the dose and route of administration need to be well conceived before starting the experiment as these may affect the bioavailability of the drug [146].

Table 1 shows the behavioral and neurochemical effects observed in pharmacological models of schizophrenia.

## 3. Genetic Models

To date, genome-wide association studies have reported various schizophrenia risk loci; however, their contribution to illness development remains unclear [147]. 

### 3.1. The Disrupted-in-Schizophrenia 1 (DISC1)

The gene is implicated in schizophrenia pathophysiology as it may be related to cognitive deficits in psychiatric disorders with cognitive symptoms [148]. The Disc1 protein is shown to regulate various neuronal processes, such as morphogenesis, maturation, migration, and synaptic integration of neuronal cells [149,150]. Some evidence indicates that *DISC1* point mutations in mice affect the development of the cerebral cortex [151]. The linkage between *DISC1* abnormalities and developing schizophrenia has been found in humans [152]; however, the *DISC1* gene and its clear connection to schizophrenia remain controversial [153].

In mouse models, it has been shown that the knockdown of *Disc1* in the hippocampus leads to deficits in recognition memory and social and anxiety behavior [148], which are characteristic symptoms of schizophrenia. Dominant-negative *Disc1* mice showed disrupted spatial and recognition memory and abnormal performance in PPI without social behavior deficits or changes in locomotor activity in [154]. In the same dominant-negative *Disc1* model, the mice were shown to exhibit deficits in social novelty preference without abnormalities in tests, such as novelty object recognition or open field. These mouse mutants showed reduced expression of GAD67, subunit 1 of NMDA receptor, and postsynaptic density 95 protein (PSD-95). These genes are critical for the pathophysiology of schizophrenia and play an important role in synaptogenesis and GABAergic and GLUergic neurotransmission. Additionally, a decreased number of neuropeptide Y (NPY) neurons was observed in the medial prefrontal cortex, anterior cingulate cortex, and orbitofrontal cortex in [155]. However, Morosawa et al. [150] indicated no changes in the frontal and hippocampal BDNF levels in *DISC1* knockout mice. In a different study, disturbances in the GABAergic system were observed in *DISC1* knockout mice, such as a decreased GABAergic neuron density in the cortex, suggesting that the GABAergic neuronal system is an important system for schizophrenia pathogenesis [156]. Niwa et al. [157] found that the knockout of *DISC1* in utero leads to disturbed postnatal DAergic maturation in the mesocortex and interneuron projection to the medial prefrontal cortex.

**Advantages**: Such a genetic model allows us to carefully study the role of the Disc-1 protein in various neuronal processes related to schizophrenia. Disturbances in brain morphology and physiology caused by mutations in the *Disc-1* gene correlate very well with the changes observed in schizophrenic subjects.

**Limitations**: The generation of this model is complicated, very expensive, and time-consuming. The genetic component of schizophrenia appears to be rather complex, and more than one gene is likely implicated in the etiopathology of schizophrenia. Thus, this model does not fully reflect the complexity of the illness but may be useful for screening new pharmacotherapeutic agents.

### 3.2. Deletion in the 22q11.2 Region Model

In 2000–4000 live births, 22q11.2 deletion syndrome occurs [158], and most genes in this locus are expressed during brain development [159].

In humans, 22q11.2 deletion is associated with several psychiatric diseases, such as psychosis, mood and anxiety disorders, and attention deficit hyperactivity disorder (ADHD) [160]. Studies have shown that subjects with 22q11.2 deletion exhibit a 25% risk of developing schizophrenia symptoms, whereas the prevalence in the general population is up to 1% [161].

Mice with deletion in the 22q11.2 region (*Df(16)A^+/−^* mice) recapitulate many features of schizophrenia, such as impairments in synaptic function and plasticity, electrophysiological properties, and both DAergic and GABAergic systems, along with disrupted connectivity of several brain regions, mainly in the frontal cortex and hippocampus [162]. Sun et al. [162] showed enhanced cell excitability, impairment in inhibitory activity, disrupted Ca^2+^ homeostasis, and changes in the expression of genes related to neuronal activity in neuronal cell cultures from *Df(16)A**^+/−^* mice. Mukai et al. [163] reported disrupted neuron morphology, axonal growth, branching of pyramidal neurons accompanied by a reduction in the strength of synaptic connections and impaired functional connectivity and working memory.

The deletion of the 22q11.2 locus in mice has been shown to produce an exacerbated response to psychostimulants expressed in increased locomotor activity and impaired sensorimotor gating measured in PPI and social memory [164,165], and these alterations are regarded as animal behaviors relevant to psychiatric conditions, including schizophrenia [166]. Mutant mice showed a decreased density of PV+ interneurons in the CA2 hippocampal region and disrupted inhibitory control of excitatory tone from the CA3 region [164], suggesting disrupted excitatory/inhibitory homeostasis of the brain.

**Advantages**: The strongest advantage of this model is the fact that the deletion in the 22q11.2 region is thought to be the strongest genetic predictor of schizophrenia development [167]. *Df(16)A**^+/−^* mice recapitulate some behaviors commonly observed in other schizophrenia models (e.g., PPI or impaired social interactions) [165,168]; thus, animals with a microdeletion of the relevant region are suitable for studying some aspects of the illness in the field (e.g., developing new therapeutics). 

**Limitations**: Similar to previous genetic models, this model is rather expensive and complicated. The results obtained from the animal model may vary depending on the size of the deletion (1.5 vs. 3.0 Mb). Moreover, using the CRISPR/Cas9 method to delete a part of the region, any unintended mutations can occur, influencing animals’ behavior or structural and molecular impairments. Behavioral data linked to learning, memory, and attention collected from studies using the 22q11.2 deletion model vary considerably [169]; therefore, the relationship between the deletion and cognitive schizophrenia-related symptoms may be difficult to interpret in terms of cause-and-effect relationship. Additionally, 22q11.2 microdeletion significantly increases the risk of developing other neurological conditions, such as parkinsonism, autism, and intellectual disability [167], rendering the results quite challenging to relate to a specific disease.

### 3.3. Dysbindin-1 Model

Genetic changes in the *DTNBP1* gene, which encodes dysbindin-1 protein, have been implicated as a gene risk factor for schizophrenia development [170]. It has been shown that dysbindin-1 is significant for brain development, appropriate spine morphology, synaptic function, and plasticity [171,172]. Furthermore, the *DTNBP1* genotype is linked to cognitive deficits [173], and genetic variations in dysbindin-1 reportedly correlate with the cognitive response to antipsychotic drugs [174]. Mice with deletion in *DTNBP1* (*sandy* mice) serve as an animal model of dysbindin-1 dysfunction as the mutation causes a significant reduction in (heterozygotes) or total loss of (homozygotes) the protein [175]. Other studies have shown that a lack of dysbindin-1 affects GLUergic, GABAergic, and DAergic transmissions [172,176], which are all important for schizophrenia pathophysiology.

*Sandy* mice display a significant reduction in the mRNA NR1 subunit of the NMDA receptor in the prefrontal cortex and abnormal NMDA receptor-driven currents [177]. Homozygous dysbindin-1-deficient mice exhibit decreased spontaneous inhibitory postsynaptic currents in the prefrontal cortex, suggesting impaired GABAergic transmission [172,178]. Additionally, the density of PV+ neurons was found to be reduced in the hippocampal formation in *sandy* mice in [172]. In DAergic pathways, dysbindin-1 deletion causes an increase in DA release and the overexpression of cell surface D2 receptors [179,180], serving as a DA-based schizophrenia model linked to positive symptoms [181].

Dysbindin-1 deletion-based models show behavioral impairments relevant to schizophrenia manifestations. In behavioral studies, mutant mice showed impaired working and spatial memory and fear conditioning processes [175,177,182]. Additionally, mutant mice exhibited social behavior impairments [183,184]. Dysbindin-1-deficient mice also present disrupted locomotor activity [175,182], which is thought to be a core feature observed in animal schizophrenia models.

Notably, postmortem studies of schizophrenic brains showed reduced levels of dysbindin-1 in regions such as the prefrontal cortex, midbrain, and hippocampus [185,186].

**Advantages**: In vitro studies have shown that dysbindin-1 null animals exhibit disturbed DAergic and GLUergic systems, which are controlled by appropriate dysbindin-1 functions [176]. Importantly, dysbindin-1 mutant mice display various behaviors relevant to schizophrenia, such as social interaction and working memory impairments [183], and thus may represent a valid model for studying these schizophrenia-related deficits. Notably, in recent years, a significant association linked to dysbindin-1 allele frequency and a trend relationship with genotype was found while comparing schizophrenia subjects and healthy controls [187], affirming the role of dysbindin-1 in developing the illness.

**Limitations**: First, dysbindin-1 knockout mice (called ‘*sandy*’ due to the color of their fur) were generated from *DBA/2J* mice, which exhibit slight, if any, PPI—a common test used to measure sensorimotor gating in schizophrenia models. Therefore, the genetic background of *sandy* mice might not represent an applicable model for PPI testing [188]. The models of single-gene mutations verify the role of a single particular molecule in the illness rather than the etiology linked to several factors. Importantly, sandy mice are not commercially available but may be delivered from *sdy/DBA* animals recovered from cryopreserved embryos. Moreover, *sandy* mice generated from neither DBA nor BL6 display anxiety behavior as measured in the elevated plus maze (EPM) or the light/dark box [181] and show lowered activity [183] in contrast to other schizophrenia animal models in which hyperlocomotion is observed and considered a core manifestation related to human positive symptoms.

### 3.4. Neurotrophic Factor Neuregulin 1 (NRG1) Model

Genes encoding the NRG1 protein and its receptor epidermal growth factor receptor 4 (ErbB4) are implicated in schizophrenia pathogenesis. *NRG1* and *ERBB4* genetic variants were found to be associated with reduced white matter integrity and volume and increased lateral ventricle volume in [189].

Importantly, homozygous null mouse embryos die at 10.5–11 d of gestation or soon after birth, whereas heterozygous individuals are viable and live to adulthood [190]. *Nrg1* mutant mice show hyperactivity, which may be reversed by antipsychotics, such as clozapine. Additionally, using the MK-801 binding technique, it has been found that mutants exhibit 16% fewer functional NMDA receptors [191]. *Nrg1* haploinsufficiency leads to a decreased number of presynaptic excitatory terminals on PV+ cortical neurons, resulting in altered inhibitory/excitatory homeostasis by reducing excitatory input to PV cells [192].

Wang et al. [193] reported that *Erbb4* deletion in adult mice results in behavioral deficits (hyperlocomotion, impaired PPI, and contextual fear conditioning) and reduced inhibitory postsynaptic currents (IPSCs) in hippocampal slices, suggesting that ErbB4 plays a critical role in GABAergic transmission. Moreover, *Erbb4* null mutant mice display lower numbers of PV+ interneurons and compromised GABA release [194]. Previous studies have shown that ErbB4 deficits impair the correct wiring of cortical neurons, affecting excitatory synapses between pyramidal neurons and PV+ cells [192,195], suggesting that the NRG1 receptor plays a critical role in excitatory/inhibitory circuit homeostasis.

Postmortem studies are consistent with preclinical investigations and have shown that the mRNA and protein expression of NRG1 and NRG1–ErbB4 signaling are disrupted in the brains of schizophrenic subjects [196,197]. It is known that dendritic spine deficits might be a significant hallmark of schizophrenia, and in mice mimicking high levels of NRG1 (cto*Nrg1*), spine development deficits have also been observed. It has been shown that cto*Nrg1* mice exhibit increased NRG1 levels by 50–100% in the forebrain, which is similar to that in schizophrenic forebrains. Therefore, cto*Nrg1* mice are a relatively better model for mimicking high levels of NRG1 in the etiopathology of schizophrenia [198].

**Advantages**: Such a genetic model allows us to carefully study the role of the NRG1 protein and its receptor ErbB4 in various neuronal processes that are disturbed in the disease. The impairments in brain morphology and physiology induced by mutations in NRG1–ErbB4 signaling correlate very well with the changes observed in people diagnosed with schizophrenia.

**Limitations**: Generating this genetically engineered model is a complicated, highly expensive, and long-lasting process. The genetic component of schizophrenia appears to be rather complex, and more than one genetic component likely changes over the schizophrenia etiopathology. Thus, this model does not fully reflect the complexity of schizophrenia development but may be useful for studying the effects of disturbed NRG1–ErbB4 signaling in the illness and searching for new therapies targeting this signaling pathway.

Table 2 shows the behavioral and neurochemical effects observed in genetic models of schizophrenia.

## 4. Neurodevelopmental Models

Neurodevelopmental models of schizophrenia are based on the hypothesis that abnormalities in the development and maturation of the prefrontal cortex and hippocampus during prenatal and perinatal life induce long-term changes in the brain that do not appear until puberty.

### 4.1. Neonatal Ventral Hippocampal Lesion (NVHL)

The NVHL model in rats has been shown to display both behavioral and neurochemical changes in the brain relevant to those observed in schizophrenia [199]. NVHL involves the surgical infusion of excitotoxin (ibotenic acid) into the hippocampus during the first week of postnatal life, which is analogous to the third trimester of human development [200]. Behavioral and molecular changes are observed during the postpubertal period, with no changes during the prepubertal period [199,201].

Numerous studies have shown that NVHL rats exhibit impaired behavior. NVHL rats display impaired sensorimotor gating [200,202,203], social interactions [203,204], attention [205], working memory [206,207], spatial memory [208,209], hyperlocomotion [204,210,211], and hypersensitivity to psychostimulants [200,212]. Additionally, NVHL animals show enhanced addictive behavior, such as increased self-administration of cocaine, nicotine, alcohol, and methamphetamine [213,214,215,216].

The NVHL model also affects the neuronal architecture of the brain, and studies suggest developmental reorganization of affected structures (e.g., the prefrontal cortex and nucleus accumbens), indicating a role for developmental changes in the functional relationship [217]. NVHL animals show neural atrophy and gliosis in the nucleus accumbens, basolateral amygdala, and both layers III and V of the prefrontal cortex manifested by reduced dendritic length, spine density, and spine population [210,217,218]. Forming and maintaining the spine density and proper morphology are closely related to BDNF/TrkB signaling [219]. In the NVHL model, the BDNF levels have been shown to be decreased in the frontal cortex and CA1/2 and dentate gyrus in the hippocampus [220,221]. Moreover, Vázquez-Roque et al. [204] showed a decreased number of neuronal cells in the prefrontal cortex of NVHL rats.

NVHL also affects the oxidative balance in brain structures by enhancing nitric oxide (NO) accumulation in the prefrontal cortex, striatum, and occipital cortex, and this effect may be reversed by haloperidol, risperidone, and aripiprazole in the prefrontal cortex [202,222]. Studies by Cabungcal et al. [223], using 8-oxo-7,8-dihydro-20-deoxyguanosine (8-oxo-dG) labelling, showed that after NVHL, rats express higher levels of oxidized DNA in developmentally compromised brains at both P21 and P61, suggesting ongoing oxidative-stress-related processes. Interestingly, these deficits may be prevented by GSH precursor N-acetyl cysteine (NAC) administration during the prepubertal stage.

Several studies have shown disturbed cytokine levels in the NVHL rat model. Increased levels of proinflammatory IL-1β and decreased levels of transforming growth factor-β1 (TGF-β1), an anti-inflammatory cytokine, have been observed; however, there are no significant changes in TNF-α or IL-6 [203]. Feleder et al. [224] used lipopolysaccharide (LPS) to develop NVHL and found a significant increase in TNF-α, interleukin-1 (IL-1), and interleukin-2 (IL-2) in brain structures, such as the nucleus accumbens, prefrontal cortex, and hippocampus, indicating long-lasting inflammatory responses in brain structures involved in schizophrenia pathophysiology.

**Advantages**: The NVHL model is relatively cheap and perfectly reflects disorders characteristic of schizophrenia. There are anatomical, histological, and physiological changes and a wide spectrum of behavioral disorders compatible with the impairments observed in patients with schizophrenia. Moreover, these changes are progressive in nature as follows: initiated during the neonatal period and developed over time, and full disease symptoms similar to those in humans are observed in adult animals. This neurodevelopmental model is valuable for identifying the processes by which a triggering event in early life may pathologically change brain development and assessing preventive interventions.

**Limitations**: The generation of an NVHL model is laborious and time-consuming as it requires surgery. Moreover, this model requires high precision as the lesion is performed in 7-day-old puppies; anesthesia and the operation of such young animals are associated with an increased risk of mortality.

### 4.2. Maternal Immune Activation (MIA)

The MIA model has been proposed as another neurodevelopmental paradigm to model schizophrenia in rodents. Prenatal exposure to viral pathogens, such as influenza, is related to an increased risk of developing schizophrenia in adulthood [225], and most research has shown increased schizophrenia cases after influenza pandemics [226,227]. The MIA model requires the administration of polyinosinic:polycytidylic acid (poly(I:C)) during the gestational stage to activate the innate immune system dependent on toll-like receptor 3 (TLR-3) signaling [225]. Some researchers also use LPS to induce an immune response, and this method also qualifies as MIA [228].

The administration of poly(I:C) may be performed during several gestational stages. The most commonly used protocol is midgestation administration (GD 12–12.5 in mice and GD 14.5–15 in rats) [229]. The MIA model provides a broad spectrum of the offspring phenotype rather than limiting the effects of poly(I:C) to a single brain region, neurotransmitter system, or developmental process [229]. Prenatal poly(I:C) treatment results in a disturbed DA system, and the offspring of MIA-treated dams exhibit an increased expression of tyrosine hydroxylase (TH), a rate-limiting enzyme in DA synthesis, in the striatum and TH-positive terminals in the ventral tegmental area (VTA) [230,231]. Additionally, after poly(I:C) administration at GD 15 in rats, increased striatal DA release was observed in an in vitro study [232]. In mice, MIA performed at GD 9 resulted in increased DA and 3,4-dihydroxyphenylacetic acid (DOPAC) concentrations in the prefrontal cortex, lateral globus pallidus, and nucleus accumbens, whereas increased homovanillic acid (HVA) levels were observed in the lateral globus pallidus and nucleus accumbens [233,234]. Ozawa et al. [235] showed increased DA turnover and decreased levels of the D2 receptor in striatal regions, whereas Mundorf et al. [236] reported lower prefrontal D2 receptor mRNA levels in MIA offspring in adolescence. Research involving 5-HT-ergic systems revealed significantly higher synthesis and release of 5-HT from the placenta 48 h after maternal poly(I:C) exposure, and fetal brains contained higher levels of 5-HT and kynurenic acid 24 or 48 h after the MIA procedure. Moreover, altered 5-HT-ergic axonal circuit formation in fetal brains was found in [237]. MacDowell and coworkers [238] reported increased protein expression of the serotonin transporter (SERT) and 5-HT_2A_ and decreased levels of 5-HT_2C_ receptors with changed levels of kynurenic pathway compounds.

Regarding most evidence highlighting the role of disturbances in the GLUergic system in schizophrenia pathogenesis, the MIA model also affects GLU transmission [239]. Rahman et al. [240] showed changes in frontal and hippocampal NMDA receptor binding and disturbed mRNA levels of the NR1 and NR2 subunits in the offspring of MIA rats. Moreover, an analysis of medial prefrontal cortex pathways to the amygdala in the MIA model provided evidence of increased GLUergic synaptic transmission in both interneurons and principal neuronal cells [241].

Evidence suggests that the MIA model provides impaired GABAergic transmission at various levels, including changes in GABA receptor expression [242]. Labouesse et al. [243] showed that prenatal immune activation in mice leads to altered promoter methylation of specific GABAergic genes, *Gad1* and *Gad2*, in the offspring. An immune challenge during late pregnancy can also decrease the mRNA and protein levels of GAD67 and GAD65, two isoforms of enzymes required for GABA synthesis, in the medial prefrontal cortex in postpubertal mice [244], whereas lower protein levels of GAD67 were found in the rat hippocampus in the MIA model within PV+ interneurons without neuronal loss in [245,246]. Canetta et al. [247] reported decreased IPSCs in medial prefrontal cortex pyramidal cells, impaired GABAergic transmission from PV+ interneurons and pyramidal cells in the medial prefrontal cortex, and lowered release probability in PV+ interneuron synapses in the medial prefrontal cortex in murine adult MIA offspring, and this diminished transmission to the medial prefrontal cortex leads to hyperactivity [248]

MIA-induced changes involve functional impairments or the loss of PV+ interneurons [247,249], which are very important for maintaining the excitatory/inhibitory brain balance [250,251]. Most evidence shows impaired behavior relevant to other animal models of schizophrenia, such as stimulant-induced increases in locomotor activity [232,235,252], deficits in PPI [253,254,255], altered anxiety [255,256], and social behavior [257,258]. Additionally, the offspring of animals immunochallenged during pregnancy exhibit impairment in working memory and cognitive flexibility [235,259,260]. These behaviors are commonly thought to be hallmarks of schizophrenia animal models and can be alleviated by antipsychotic administration [261,262], supporting the disease relevance of the MIA model [228].

**Advantages**: The generation of the MIA phenotype is a relatively simple, inexpensive, and not prolonged process as it is based only on intraperitoneal injection of poly(I:C) on GD 12–12.5 in mice or GD 14.5–15 in rats. This model substantially reflects disorders characteristic of schizophrenia. The MIA model generates anatomical, histological, and physiological changes and a wide spectrum of behavioral disorders relevant to the changes observed in schizophrenic subjects. These changes are progressive as they initiate during the prenatal stage of life and are then observed during the postpubertal period, which may resemble the course of developing the illness.

**Limitations**: The days of gestation should be accurately determined, and after the intraperitoneal injection, there is a risk of miscarriage. Moreover, the model requires more time as the offspring of immunochallenged rats must reach the postpubertal stage.

### 4.3. Methylazoxymethanol Acetate (MAM) Model

MAM is one of the most popular neurodevelopmental models of schizophrenia. This model is based on the prenatal administration of MAM on embryonic day 17, which causes disruption of embryonic brain development [263]. This model allows us to understand schizophrenia pathology and the mechanism of action of antipsychotic drugs and examine potential novel treatments [264,265]. After using the MAM model, the deficits that appeared in the adult rodents paralleled those observed in schizophrenic patients. MAM is a DNA methylating agent that induces augmentation in DA transmission as follows: MAM produces an increase in DA release in forebrain structures correlated with an elevation in the number of spontaneously active DA neurons [266]. In the MAM model, increased DA neuron function causes aberrant hippocampal activity, which is consistent with disorders observed in human schizophrenia patients. MAM administration to pregnant rats induces a reduction in cortical thickness and simultaneous elevation in neuronal density (mainly in the medial prefrontal cortex and hippocampus). Moreover, decreased PV expression was observed in both brain structures, and deficits in gamma oscillatory activity and alterations in H3 histone methylation were observed in the medial prefrontal cortex in [267,268]. The excitation–inhibition balance in the prefrontal cortex is associated with physiological DA–GLU–GABA interactions, and disturbances in the development of these interactions cause dramatic functional changes and behavioral deficits in adult rats. These neuroanatomical changes disrupt corticocortical synaptic transmission, leading to striatal and hippocampal hyperdopaminergia and altered GLUergic activity in the hippocampus. The administration of MAM to pregnant rats also causes several behavioral disturbances observed in adult offspring, such as cognitive dysfunction, reduced social interaction, sensorimotor gating deficits, and perseverative responding [269]. In addition, in [270,271], the MAM-treated rats showed the following abnormal stress response: they emitted significantly more vocalizations at a higher rate and had an increased anxiety response, greater freezing after footshock, higher levels of anxiety-like behavior in the elevated plus-maze, and social interaction deficits. Interesting research has emerged in recent years, indicating that the enriched environment (EE) used in models of schizophrenia reverses behavioral changes, such as hyperactivity, sensorimotor gating deficits, memory impairment, and social interaction deficits [272,273]. These behavioral effects of EE are related to the prevention of MAM-induced hyperactivity of pyramidal neuron in the rat’s hippocampus, as well as the DA hyperactivity in the VTA [273].

**Advantages**: As researchers maintain, the MAM allows one to understand schizophrenia pathology and the mechanism of action of antipsychotic drugs and examine potential novel treatments [264,265]. The generation of the MAM phenotype is a relatively simple, inexpensive, and not time-consuming process and consists only of intraperitoneal injection on embryonic day 17. This model reflects disorders characteristic of schizophrenia as anatomical, histological, and physiological changes are observed in MAM animals. Additionally, a wide spectrum of behavioral disorders compatible with the changes observed in patients with schizophrenia are present in MAM rats. Similar to the MIA model, pathological changes are initiated during the prenatal stage and, therefore, affect brain development, and abnormal behavior and functional/molecular changes are observed in adult life.

**Limitations**: MAM is a toxic and carcinogenic compound; thus, special precautions should be taken as follows: Animals should be isolated from other animals, and the researcher should wear protective clothing (gloves, masks, glasses, etc.). All gloves, pipette tips, and empty MAM vials should be disposed of in a safe manner. Moreover, the day of gestation should be accurately determined, and after the intraperitoneal injection, there is a risk of miscarriage.

Table 3 shows the behavioral and neurochemical effects observed in neurodevelopmental models of schizophrenia.

## 5. Translational Values of the Animal Models

Animal models are indeed valuable tools in science to model human psychiatric disorders, especially in improving knowledge of the neurobiological basis of medical conditions and developing new effective treatments. However, they meet several limitations regarding the interspecies differences between rodents or zebrafish and humans. Mental illnesses are extremely difficult to model in laboratory animals, especially due to the complexity of the brain structure and inscrutable human mind. 

It is commonly thought that schizophrenia animal models need to meet several conditions to be reliable and to have translational value. According to an excellent review by Jones and coworkers [274], these are: (1) symptoms homology—the model is expected to mirror core schizophrenia behavioral symptoms: positive, negative, and cognitive ones; (2) construct—the model needs to replicate the pathology of neurochemical/structural changes, such as DA and GLU system dysregulations and synaptic deficits; and (3) predictive—the model needs to show the effectiveness of commonly used antipsychotics with well-known mechanisms of action and/or the lack of effect in newly screened agents. 

To the best of our knowledge, most of the models do not meet this triad of requirements in a perfect manner. The most challenging is to replicate the schizophrenic mind, as the thoughts or verbal processes, such as learning, are unique to humans [275] and are unmeasurable in, for example, rodents. Additionally, modeled animals are unable to self-report hallucinations, alogia, and other schizophrenia-related features [275]. Therefore, animal models of disorders are unable to reflect such symptoms, and we cannot possess knowledge of the effectiveness of the tested pharmacotherapies in attenuating these symptoms in animals. 

In the laboratory, animal models can recapitulate only some of the schizophrenia-related features. The most challenging is to combine as many as possible predisposing factors to obtain the best picture of behavioral, anatomical, and neurochemical changes in modeled animals. Here, worth mentioning are the genetic models, in which one single gene is knocked out or mutated to resemble these genetic impairments present in schizophrenic subjects. The genetic background of schizophrenia is more complex than that, and apart from the genetic components, environmental factors play a significant role in developing the illness in human subjects [276]. Similarly, the pharmacological models affect one neurotransmitter system when changes in brain neurochemistry in schizophrenia is more complex. Indeed, every system that interacts with others, such as GLUergic and DAergic, can be controlled by each other [277], and both of them are observed to be disturbed in schizophrenia, but it does not enlighten the human pathophysiology of the development of the illness. This complexity of the known neurobiological basis of schizophrenia raises the following questions: which of the observed changes are the results of another, and which of them are the main triggering factor to develop the illness? Thus, none of the mentioned models fully mirror human schizophrenia conditions, and individual ones are developed to reflect different patterns of symptoms. However, without the contribution of laboratory schizophrenia models, none of the pharmacotherapies could have been developed or improved.

## 6. Summary

Schizophrenia is one of the human neuropsychiatric diseases with a very complex etiology characterized by a wide spectrum of behavioral symptoms and neurochemical changes. Therefore, modeling this disease is extremely difficult and challenging. Nonetheless, schizophrenia animal models are widely used in preclinical research for constant upgrading of schizophrenia pharmacotherapy and our understanding of the disorder. We can distinguish among pharmacological, genetic, and neurodevelopmental models offering various neuroanatomical disorders and a different spectrum of symptoms of schizophrenia. Table 4 shows a comparison of animal schizophrenia models.

Pharmacological models, compared with other types of animal schizophrenia models, are relatively a very simple, inexpensive, and not time-consuming process. They reflect most of the behavioral disturbances observed in humans, such as positive and negative symptoms, and memory and learning disorders. Unfortunately, pharmacological manipulations do mostly resemble acute psychosis rather than a permanent state of the illness. Notably, compounds are used to study schizophrenia-related behavior and molecular changes after acute or subchronic drug administration, whereas schizophrenia is a chronic disease with manifestations starting at puberty or shortly after this time. Therefore, pharmacological manipulations do not mimic the pathogenesis of schizophrenia efficiently. These manipulations may resemble the symptoms; however, their ability to model the course of the development of the illness is rather limited.

Particular genetic models focus on mutations/damage within a single gene that correlates with the symptoms of schizophrenia. They are an excellent tool for studying the role of individual proteins in the development of this disease and the appearance of behavioral schizophrenia symptoms. Unfortunately, genetically engineered models are complicated, highly expensive, and long-lasting processes. The genetic component of schizophrenia appears to be rather complex, and more than one genetic component likely changes over the schizophrenia etiopathology. In addition, the development of schizophrenia can be influenced by the environment.

Neurodevelopmental models seem to offer the most advantages compared with other types of animal models of schizophrenia. Especially in these models, behavioral and molecular changes are observed during the postpubertal period, with no changes during the prepubertal period. Thus, these changes are progressive in nature as follows: they are initiated during the neonatal period and developed over time, and full disease symptoms similar to those in humans are observed in adult animals. In these models anatomical, histological, and physiological changes are observed simultaneously with a wide spectrum of behavioral disorders, compatible with the impairments observed in patients. Therefore, neurodevelopmental models are valuable for identifying the processes by which a triggering event in early life may pathologically change brain development and assessing preventive interventions. The greatest limitations of these models are necessity of precisely defined days of drug administration in pregnant females or in pups of rats and a long period of waiting for the effects of drug administration, which is associated with long-term care for a given group of animals and increasing costs of their maintenance.

Summing up, all animal models of schizophrenia have advantages and drawbacks. The use of these models mainly aims to gain a better understanding of the etiopathology of the disease and the development of new antipsychotic drugs. Before starting the experiments, the appropriate animal model should be chosen depending on the available budget, time, and desired effect of the intervention, which should be appropriate regarding, for example, the examined therapeutic compounds. Most importantly, when focusing on the advantages of individual animal models of schizophrenia, one should not forget their limitations as animal models never entirely reflect the complexity of human illnesses.

## Figures and Tables

**Figure 1 ijms-23-05968-f001:**
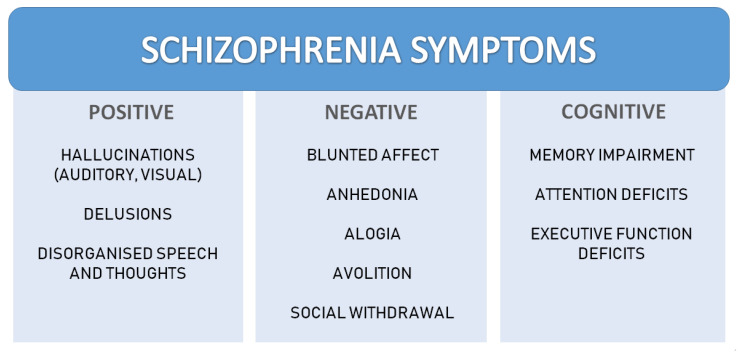
Schizophrenia symptoms.

**Table 1 ijms-23-05968-t001:** Several examples of pharmacological intervention and its behavioral and molecular effects.

Species	Pharmacological Intervention	Behavioral Effects	Molecular Effect	Reference
Mice	acute AMPH 1.25 mg/kg *i.p.*	↑ locomotor activity ↓ working memory performance ↓ sociability ↑ stereotypy behavior	n.d.	[15]
Rat	5 doses of AMPH 2.5 mg/kg *s.c.*	↑ latency time in water maze task ↓ alternations in Y-maze task ↑ investigation time in SI	↑ MDA ↑ TNF-α	[16]
Rat	9 doses of AMPH, increasing dose 1 mg/kg–3 mg/kg, *i.p.*	↑ locomotor activity ↑ sensitivity to stimulants PPI deficits LI deficits	↓ binding to D2 receptors	[17,18]
*Danio rerio* (zebrafish)	acute AMPH 0.625 2.5 10 mg/L	no changes in locomotor activity no changes in social interaction	↑ MDA	[23]
Rat	6-OHDA intracerebral (mPFC) injection	impaired contextual fear conditioning ↓ sociability	↓ 5-HT level in mPFC ↑ ratio DOPAC/DA in mPFC ↓ HVA level in mPFC ↑ DA level in NAcc	[30]
Rat/mice	DOI 2.5 mg/kg *s.c.* or 0.5 mg/kg *s.c.*	head-twitch response PPI disruption	↑ GLU release in mPFC	[36,40]
Mice	PCP 2.5 mg/kg for 15 days and 10 mg/kg at 16 day, *s.c.*	↑ locomotor activity ↑ anxiety behavior ↓ memory performance SI deficits PPI disruption at 80 dB	n.d.	[49]
Mice	acute PCP (10 mg/kg, *i.p.*)	↑ locomotor activity	↑ efflux of ACh, DA, NA, 5-HT, GLU in mPFC, and dSTR	[53]
Rat	subchronic PCP (2 mg/kg *i.p.* 2 x day for 7 days)	↓ performance in attentional set-shifting task in ♀↑	↓ BDNF level in mPFC, motor cortex, OFC, OB, RSP cortex, FCX, parietal cortex, CA1, central, lateral, basolateral amygdala (in ♀)	[57]
Mice	subchronic PCP (10 mg/kg, *s.c.* for 10 days)	locomotor activity ↑ immobility time in FST ↓ memory performance	In PFC: ↑ mRNA of IL-1β, TNF-α, IL-6, COX-2, and iNOS ↑ protein level of TNF- α and IL-1 β ↑ phosphorylation of p38, p65, IκBα, ERK1/2 ↑ MDA, ↓ activity of SOD, CAT, GSH-Px ↓ GSH level	[60]
Rat	acute KET (20 mg/kg *i.p.*)	↓ memory performance	↑ NA efflux in STR ↓ DOPAC and HVA in STR	[72]
Mice	subchronic KET (20 mg/kg *i.p.* for 7 or 14 days)	↑ locomotor activity ↓ sociability ↓ spatial and recognition memory performance	↓ SOD, CAT and AChE activity ↓ GSH level ↑ MDA level ↑ AChE ↑ nitrite	[74]
Rat	subchronic KET (25 mg/kg, *i.p.* for 7 days)	↑ locomotor activity ↓ no. of social contacts PPI deficits	In FCX: ↑ lipid hydroperoxide ↑ 4-HNE level ↑ 8-isoprostane ↑ SOD, CAT activity (and in HIP) ↓ BDNF and NGF levels	[85]
Mice	subchronic KET (20 mg/kg *i.p.* for 14 days)	PPI deficits ↓ memory performance ↓ social preference	in PFC, HIP, STR: ↑ MPO activity ↑ MDA level ↓ GSH level ↑ IL-4 and IL-6 levels (in HIP)	[87]
Rat	acute MK-801 (0.1 mg/kg *i.p.*)	↓ spatial and recognition memory performance	↓ p-TrkB	[98]
Mice	subchronic MK-801 (0.1 mg/kg *i.p.* for 7 days)	↓ sociability ↑ locomotor activity ↑ anxiety behavior MWM deficits	in HIP: impaired LTP ↓ PSD-95 and SYP levels ↑ microglia ↑ IL-6 and IL-1β ↑ COX-2 and iNOS	[103]
Mice	subchronic MK-801 (1 mg/kg *i.p.* for 14 days)	↑ locomotor activity ↓ sociability PPI deficits	n.d.	[104]
Rat	acute MK-801 (0.3 mg/kg *i.p.* in FM; 0.1 mg/kg *s.c.* in SI)	↓ sociability FM deficits	↑ NA in FCX ↓ DA in FCX ↑ DOPAC in HIP	[106]

Abbreviations: ↑: increased/enhanced; ↓: decreased/weakened; n.d.: no data; 4-HNE: 4-hydroxynonenal; 5-HT: serotonin; 6-OHDA: 6-hydroxydopamine; ACh: acetylcholine; AChE: acetylcholinesterase; AMPH: amphetamine; BDNF: brain-derived neurotrophic factor; CAT: catalase; COX-2: cyclooxygenase-2; DA: dopamine; DOI: 2,5-dimethoxy-4-iodoamphetamine; DOPAC: 3,4-dihydroxyphenylacetic acid; dSTR: dorsal striatum; ERK 1/2: extracellular signal-regulated kinase 1/2; FCX: frontal cortex; FST: forced swimming test; GLU: glutamate; GSH: glutathione; GSH-Px: glutathione peroxidase; HIP: hippocampus; HVA: homovanillic acid; i.p.: intraperitoneal; IL-1β: interleukin-1β; IL-4: interleukin-4; IL-6: interleukin-6; iNOS: inducible nitric oxide synthase; IκBα: nuclear factor of kappa light polypeptide gene enhancer in B-cells inhibitor alpha; KET: ketamine; LI: latent inhibition; LTP: long-term potentiation; MDA: malondialdehyde; mPFC: medial prefrontal cortex; MPO: myeloperoxidase; NA: noradrenaline; NAcc: nucleus accumbens; NGF: nerve growth factor; OB: olfactory bulb; OFC: orbitofrontal cortex; PCP: phencyclidine; PFC: prefrontal cortex; PPI: prepulse inhibition; PSD-95: postsynaptic density protein 95; p-TrkB: Phosphorylated tropomyosin receptor kinase B; RSP: retrosplenial; s.c.: subcutaneous; SI: social interaction; SOD: superoxide dismutase; STR: striatum; SYP: synaptophysin; TNF-α: tumor necrosis factor alpha.

**Table 2 ijms-23-05968-t002:** Several examples of genetic mutations used to model schizophrenia and its behavioral and molecular effects.

Species	Type of Mutation	Behavioral Effects	Molecular Effects	Reference
Mice	dominant negative DISC1	↓ social novelty preference	↓ TrkB in PFC ↓ CB1R in FCX (♀) and HIP (♂)	[155]
Mice	DISC1 k.o. in utero in PFC	PPI deficits hypersensitivity to stimulants ↓ working memory performance	dendritic abnormalities impaired electrophysiology of neuronal cells ↓ extracellular DA level in mPFC ↓ TH level in mPFC ↓ PV level in mPFC ↓ EPSPs of pyramidal cells	[157]
Mice	22q11.2 deletion	PPI deficits ↓ spatial memory performance ↓ social memory ↑ locomotor activity fear memory deficits	↓ synaptic plasticity ↓ spine density in mPFC	[165]
Mice	22q11.2 deletion (*Df(h22q11)/+*)	PPI deficits hypersensitivity to stimulants	↑ DOPAC in PFC and dSTR ↓ NeuN in PFC ↑ GluR1 in dSTR ↑ LDAEPs	[168]
Mice	dysbindin-1 loss *(sdy/sdy)*	↑ locomotor activity	n.d.	[175]
Mice	dysbindin-1 loss Dys-/dys-	↓ working memory performance PPI deficits ↑ locomotor activity	impaired firing of pyramidal cells ↓ CaMKII in PFC ↓ CaMKKβ in PFC	[176]
Mice	dysbindin-1 loss Dys-/dys-	↑ locomotor activity	↑ D2R in cortical neurons ↑ recycling of internalized D2R ↓ fast-spiking neuron excitability in PFC and STR ↓ GABAergic transmission in PFC	[178]
Mice	Sdy/Sdy	↑ sensitivity to stimulants ↓ working memory performance impaired fear conditioning observed hypoalgesia	n.d.	[182]
Mice	NRG1 hypomorphs and ErbB4 hypomorphs	↑ locomotor activity PPI deficits (in NRG1 hypomorphs)	↓ NMDAR functionality in NRG1 hypomorphs	[191]
Mice	ErbB4−/−	n.d.	↓ PV+ cells in hippocampus ↓ GAD67+ cells in hippocampus ↓ n-NOS+ cells in hippocampus	[194]

Abbreviations: ↑: increased/enhanced; ↓: decreased/weakened; n.d.: no data; ♀: females; ♂: males; CaMKII: calcium/calmodulin-dependent protein kinase II; CaMKKβ: calcium/calmodulin-dependent protein kinase beta; CB1R: cannabinoid receptor 1; D2R: dopamine receptor D2; DA: dopamine; DOPAC: 3,4-dihydroxyphenylacetic acid; dSTR: dorsal striatum; EPSPs: excitatory postsynaptic potentials; FCX: frontal cortex; GABA: gamma-aminobutyric acid; GAD67: glutamate decarboxylase 67; GluR1: AMPAR subunit glutamate receptor 1; HIP: hippocampus; LDAEPs: loudness dependence of auditory evoked potentials; mPFC: medial prefrontal cortex; NeuN: neuronal nuclear protein; NMDAR: N-methyl-D-aspartate receptor; nNOS: neuronal nitric oxide synthase; NRG-1: neuregulin-1; PFC: prefrontal cortex; PPI: prepulse inhibition; PV: parvalbumin; TH: tyrosine hydroxylase; TrkB: tropomyosin receptor kinase B.

**Table 3 ijms-23-05968-t003:** Several examples of developmental interventions used to model schizophrenia and its behavioral and molecular effects.

Species	Type of Intervention	Behavioral Effects	Molecular Effects	Reference
Rat	NVHL (0.3 μL ibotenic acid)	↑ locomotor activity ↓ social memory performance PPI deficits	↓ TGF-β1 (PD15 and PD60) in mPFC ↑ IL-1β (PD15) in mPFC ↓ spine density	[185]
Rat	NVHL (0.3 μL ibotenic acid)	↑ locomotor activity ↑ sensitivity to stimulants ↓ sociability PPI deficits	↓ spine density in PFC ↓ spine length in PFC and NAcc ↓ no. of neurons in PFC ↑ TH+ cells in NAcc shell	[186]
Rat	NVHL (0.3 μL ibotenic acid)	↑ locomotor activity ↓ sociability	↓ spine length in PFC ↓ spine density in PFC ↓ no. of neurons in PFC ↓ TrkB-FL in PFC ↓ PI3K in PFC ↓ p-ERK/ERK level in PFC ↑ COX-2 ↓ PPARƔ ↑ NO2- ↑ MDA	[192]
Rat	MIA–poly(I:C) (4.0 mg/kg, *i.v.*)	deficits in latent inhibition ↑ sensitivity to stimulants	↑ cell loss in HIP ↑ KCl-induced DA release in STR	[214]
Mice	MIA–poly(I:C) (5.0 mg/kg, *i.p.*)	↑ thigmotaxis PPI deficits ↓ memory performance ↑ sensitivity to stimulants	↑ DOPAC in STR ↑ HVA in STR ↑ ratio of metabolites of DA/DA ↓ binding to D2R in STR	[217]
Rat	MIA–poly(I:C) (4.0 mg/kg, *i.v.*)	PPI deficits ↑ locomotor activity ↓ memory performance in males	n.d.	[234]
Rat	MAM (20 mg/kg, *i.p.*)	↑ anxiety behavior ↑ sensitivity to stimulants	↑ activity of DA neurons in VTA	[251]
Rat	MAM (25 mg/kg, *i.p.*)	↑ locomotor activity ↑ sensitivity to stimulants ↓ spatial memory performance ↓ sociability PPI deficits	changes in brain structure ↓ surface area of PFC and HIP	[252]

Abbreviations: ↑ increased/enhanced; ↓ decreased/weakened; n.d. no data; COX-2: cyclooxygenase-2; D2R: dopamine receptor D2; DA: dopamine; ERK: extracellular signal-regulated kinase; HIP: hippocampus; IL-1β: interleukin-1β; MAM: methylazoxymethanol acetate; MDA: malondialdehyde; MIA: maternal immune activation; mPFC: medial prefrontal cortex; NAcc: nucleus accumbens; NO2−: nitrites; NVHL: neonatal ventral hippocampal lesion; PD: posnatal day; p-ERK: phosphorylated extracellular signal-regulated kinase; PFC: prefrontal cortex; PI3K: phosphoinositide 3-kinase; poly(I:C): polyinosinic:polycytidylic acid; PPARƔ: peroxisome proliferator-activated receptor gamma; PPI: prepulse inhibition; STR: striatum; TGF-β1: transforming growth factor β1; TH: tyrosine hydroxylase; TrkB-FL: full-length isoform of tropomyosin receptor kinase B; VTA: ventral tegmental area.

**Table 4 ijms-23-05968-t004:** Comparison of animal models of schizophrenia.

Model	Positive Symptoms	Negative Symptoms	Cognitive Symptoms	Progress of Schizophrenia	Role of Proteins	Time-Consuming	The Degree of Difficulty in Making the Model	Expense
Amphetamine model	+	−	+	−	−	−	low	low
6-OHDA model	−	+	+	+	−	+	high	low
DOI model	+	−	+	−	−	−	low	low
PCP model	+	+	+	−	−	−	low	low
Ketamine model	+	+	+	−	−	−	low	low
MK-801 model	+	+	+	−	−	−	low	low
DISC-1 model	−	+	+	−	+	+	high	high
22q11.2 deletion	+	+	+	−	+	+	high	high
Dysbindin-1 model	−	+	+	−	+	+	high	high
NGR1 model	+	−	+	−	+	+	high	high
NVHL model	+	+	+	+	−	+	high	low
MIA model	+	+	+	+	−	+	low	low
MAM model	+	+	+	+	−	+	low	low

## Data Availability

Not applicable.

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
