# Peer review of "Advantages and Limitations of Animal Schizophrenia Models"

_ijms, 2022, doi:10.3390/ijms23115968_

Round 1

Reviewer 1 Report

Manuscript is result of a lot of work. In spite of it, presents important limits.

Main concerns

1. Authors in Introduction some provide information concerning the currently used antipsychotics and emphasize their limits.

2. Dopamine (DA) and schizophrenia

Authors correctly assessed the role of amphetamine in schizophrenia but did not consider the role of DA agonists such as apomorphine and quinpirole in this context. The rational of this choice remain unclear.

Authors describe the hyperfunction DA model of schizophrenia which is related to positive symptoms of this psychiatric disorder but overlooked the hypofunction DA model which deals with the negative symptoms and cognitive deficits of schizophrenia (e.g., Front Neurosci 14, 632, 2020).

DA-related motivational models which resemble negative symptoms are well documented.

3. Further, the DOI-induced head switch response which is a serotonergic tool for schizophrenia was not discussed at all in the pharmacological models.

4. PCP

I fully disagree with the conclusion that PCP is of lower value as a glutamatergic agent compared to ketamine or to MK-801. PCP was the first compound which was tested and found effective in the social interaction test, a procedure with high translational value for the negative symptoms of schizophrenia.

5. A critical comparisons of all models is missed. Their translational value was not commented.

Minor concerns

1. Please provide some references regarding the general concepts of schizophrenia in Introduction.

Reviewer 2 Report

This review manuscript provides a comprehensive review of the current filed of animal models for schizophrenia. The authors have done an excellent work in reviewing the current models and, therefore, this review will be very useful for the active researchers in this field. I have two suggestions for the authors as follows:

  1. Section 2B. Glutamatergic hypofunction models. The advantages and limitation sections are missing important information. The acute and subchronic PCP model recreate a wide array of schizophrenia symptoms. Both models produce positive, negative, and cognitive symptoms of the disease. Please describe this in the advantage section. The limitation section mentions that these two models only produce symptoms related to psychosis state. However, there are hundreds of publications showing all three category of symptoms in both acute and subchronic models. Please describe these issue. Also, please provide separate description of acute and subchronic advantages and limitations These two models are widely used and a detailed description of the two models will be very useful for the field.
  2. Section 4C. MAM Model. recent data from Anthony Grace lab and others show that environmental enrichment provided to born neonatal rats during development reverses the behavioral and cognitive deficits produced by gestational MAM treatment. This is a very important new development for MAM model which has only been recently published. It is very important to researchers in the field to know about the effect of environmental enrichment on the MAM model. Please describe this phenomena in detail. It is also true the enrichment may have the same type of effects on other neurodevelopmental models. Please explain.   

Author Response

Dear Reviewer,

First of all, I would like to apologize for the situation which was not our fault. On May 4 we received 2 reviews of our manuscript (without Yours), while your review in the system appeared on May 9 in the evening. Our replies to 2 reviews were posted on May 9 at noon! Now there are 3 reviews which is a complete surprise for us. We did not respond to Your comments because the Editor did not share your review with us. Now we have revised our manuscript in line with Your comments as well.

Reviewer 3 Report

Worth while effort comparing putative animal models of schizophrenia, and authors have done a very good job. It is important to stress that schizophrenia is essentially a human illness and its clinical manifestation is highly heterogenous. With this limitation, only certain aspects could be modelled in animals. Therefore, strength of each putative model described should be evaluated in the context of which aspect of the disease is being modelled, and limitations should be discussed accordingly for the attempted aspect of the illness. In several places, words are missing and description is not accurate (e.g., page 3, section 2A: the reference cited is not describing prefrontal cortex - should be corrected). Manuscript should be revised carefully and please describe exact interpretation/conclusions of cited references. Tables are excellent and extremely useful. 

Round 2

Reviewer 1 Report

Manuscript has substantially been improved respect to its former version.

Some issues however, still need a further consideration.

  1. Authors should include a sentence in which the inefficacy of currently used antipsychotics (40% of patients do not respond to treatment) and the first generation antipsychotics display strong side effects, including parkinsonism, while the atypical antipsychotics are related to hyperlipidemia etc. This sentence should follow  "Currently used antipsychotics..."
  2. Authors should explain their selection regarding the hyperfunction DA model  (amphetamine vs. apomorphine).

Reviewer 2 Report

Unfortunately, the authors have not responded to the two issues that I have raised about this review manuscript.

Reviewer 3 Report

Thanks for considering my suggestions. I am happy with the content and flow of the manuscript. 

Author Response

Thank you very much for approving our manuscript.

Round 3

Reviewer 2 Report

I thank the authors for incorporating the requested information.